

# A Network Model for Characterizing Brine Channels in Sea Ice

Ross M. Lieblappen[1,2], Deip D. Kumar[3], Scott D. Pauls[4], and Rachel W. Obbard[1]

[1]14 Engineering Drive, Thayer School of Engineering, Dartmouth College, Hanover, NH, USA
[2]124 Admin Drive, Vermont Technical College, Randolph Center, VT, USA
[3]6211 Sudikoff Lab, Department of Computer Science, Dartmouth College, Hanover, NH, USA
[4]27 N. Main Street, Department of Mathematics, Dartmouth College, Hanover, NH, USA

*Correspondence to:* Ross M. Lieblappen (Ross.Lieb-Lappen@vtc.vsc.edu)

**Abstract.** The brine network in sea ice is a complex labyrinth whose precise microstructure is critical in governing the movement of brine and gas between the ocean and the sea ice surface. Recent advances in three-dimensional imaging using x-ray micro-computed tomography have enabled the visualization and quantification of the brine network morphology and variability. Using imaging of first-year sea ice samples at in-situ temperatures, we create a new mathematical network model to
characterize the topology and connectivity of the brine channels. This model provides a statistical framework where we can characterize the pore networks via two parameters, depth and temperature, for use in dynamical sea ice models. Our approach advances the quantification of brine connectivity in sea ice, which can help investigations of bulk physical properties, such as fluid permeability, that are key in both global and regional sea ice models.

## 1 Introduction

The detailed microstructure of sea ice is critical in both governing the movement of fluid between the ocean and the sea ice surface and controlling processes such as ice growth and decay (Thomas and Dieckmann, 2009; Petrich et al., 2006). Its complex pore structure influences many of the bulk thermal and electric properties of sea ice. The permeability is of primary interest to a wide range of disciplines (e.g., biology and atmospheric chemistry) as it controls fluid flow through sea ice. The "Rule of Fives" provides a guideline for describing the percolation threshold in first-year columnar sea ice. Specifically, the ice
becomes permeable to fluid transport at brine volume fractions greater than $5\%$, which are found in ice at about $-5\,^\circ\text{C}$ with a salinity of about 5 parts per thousand (Golden et al., 1998). Although this rule of thumb is helpful in describing and modeling basic phenomenon, it does not fully capture the spatially and temporally evolving details of the sea ice microstructure. Here we provide a more topologically complete characterization of sea ice pore structure.

    Previous research has recognized the importance of thermally activated percolation thresholds (e.g., Cox and Weeks, 1975;
Weeks and Ackley, 1982; Golden et al., 1998; Thomas and Dieckmann, 2009). Pringle et al. (2009) studied single-crystal laboratory-grown ice using $\mu$CT to examine the thermal evolution of brine inclusions. They found that brine volume fraction and pore space structure depend upon temperature, with a percolation threshold observed at $4.6 \pm 0.7\%$. However, one expects natural polycrystalline ice to have a higher threshold as pathways are sensitive to grain boundaries, flaws, and a certain degree of horizontal transport (Pringle et al., 2009). Since different growth rates in natural sea ice produces different average spacing



between brine layers, there is also the potential for varying percolation thresholds and degree of connectivity with depth (Nakawo and Sinha, 1984; Petrich et al., 2006; Pringle et al., 2009).

In this manuscript, we develop a methodology for describing the morphology and variability of brine networks in a vertical column of first-year sea ice. We construct a network model of the pore structure of sea ice and use topological techniques to

characterize this brine network. This yields a set of network statistics that chacterizes channels of different depths and temperature, which we can later use to inform more sophisticated models of sea ice. Our framework enables us to statistically replicate the pore structure of sea ice at different depths and temperatures. Future applications include refining under what conditions the "Rule of Fives" applies, predicting bulk physical properties such as heat transfer and fluid permeability, and improving the ability to describe processes such as brine drainage and desalination. This approach provides advances in quantifying the brine

connectivity in sea ice, which we can then incorporate into both global and regional sea ice models.

## 2   Methods

This work will focus on two of the ice cores extracted from different locations in the Ross Sea, Antarctica during a October - November 2012 field campaign. The 1.78 m Butter Point ice core was collected at $77°35.133'$ S and $164°48.222'$ E and had a temperature gradient ranging from $-16.1\,°C$ at the top to $-2.5\,°C$ at the bottom. The 1.89 m Iceberg Site ice core was located

at $77°7.131'$ S and $164°6.031'$ E and had a temperature gradient ranging from $-17.7\,°C$ at the top to $-2.3\,°C$ at the bottom. Cubic sub-samples measuring 1 cm on edge were taken from each core at 10-cm intervals. We used x-ray micro-computed tomography ($\mu$CT) to image each sub-sample following the protocol developed by Lieb-Lappen et al. (2017). We scanned each sub-sample from these two cores at in-situ temperatures using a Peltier cooling stage attached to our Skyscan 1172 $\mu$CT scanner, and analyzed the three-dimensional morphological data.

Using the methods described in detail in Lieb-Lappen et al. (2017), we converted the binarized images depicting the brine phase into a network model. A network is a collection of nodes and edges connecting the various nodes (Newman, 2011). Scientists and mathematicians use networks in a wide variety of fields to create simple representations of real world systems, from which they can capture critical features and model particular behavior of the system. For example, soil scientists and geologists use network models of pore space to study connectivity and permeability (e.g., Pierret et al., 2002; Delerue et al.,

2003). Some of the common algorithms used in creating these networks include maximal ball, medial axis, and flow velocity methods (Dwyer, 1993; Silin and Patzek, 2006; Dong et al., 2008).

Here we use a version of maximal ball where we define the nodes of the network as a collection of points in three-dimensional space $\{p_i = (x_i^*, y_i^*, z_i^*) \in \mathbb{R}^3\}$, with each point assigned a radius/throat size $\{r_i\}$ (Dwyer, 1993; Silin and Patzek, 2006). By setting the network in the spatial setting of $\mathbb{R}^3$ with $z$ representing the vertical direction, we can examine questions of brine

movement upwards and downwards through the sea ice. We identify the nodes of the network $\{p_i\}$ by examining the binarized horizontal slices and locating the centroids of each two-dimensional connected component in each slice. The centroids are a reasonable approximation since viewed in two-dimensional slices, brine inclusions are primarily convex polygons with the centroid located inside the connected component (Heijmans and Roerdink, 1998). We note that each node $p_i$ located at



$(x_i^*, y_i^*, z_i^*)$ has a collection of points $\{x_i^j, y_i^j, z_i^j\}$ (where $z_i^j = z_i^*$ for all $j$) that collectively define the two-dimensional connected component of the brine channel at this $z$-value. We fit an ellipse to each connected component and we define the length of the semi-minor axis to be the throat size $r_i$ assigned to the respective node $p_i$. Thus, each brine pocket is summarized by the four-dimensional vector $(x_i^*, y_i^*, z_i^*, r_i)$. This definition captures both the location and the size of the brine phase at any point in the sea ice.

Over the collection of brine pockets, we calculate a series of probability distributions that capture how the brine channels evolve as we move vertically through the sample. For each brine pocket $p_{i1} = (x_{i1}^*, y_{i1}^*, z_{i1}^*, r_{i1})$, we look at the overlap between it and all the brine pockets at height $z - 1$. For two nodes $p_{i1}$ and $p_{i2}$ located in adjacent horizontal slices $z_{i1}^*$ and $z_{i2}^*$ (where $z_{i2}^* = z_{i1}^* - 1$), we place an edge between the two if $\{(x_{i1}^j, y_{i1}^j)\} \cap \{(x_{i2}^j, y_{i2}^j)\} \neq \emptyset$. If there are no intersections, the brine channel has terminated and we say the brine pocket $p_{i1}$ has *died*. If there is a single intersection, we say the brine pocket *remains*. If there are multiple intersections for $p_{i1}$, we say the brine pocket $p_{i1}$ *splits*, and record all such edges to the various adjacent nodes. Last, if more than one pocket at height $z$ overlaps with one pocket at height $z - 1$, we say those pockets *join* and record all incident edges. In cases where a split and join happen simultaneously, we record both (in our samples, this does not happen very often). Thus, we are able to depict the vertical connectivity of the brine phase through our definition of edges, while horizontal connectivity is captured within the definition of the node itself and we note that there are no horizontal edges. Since brine channels are primarily vertically oriented with branches splitting both upwards and downwards, this network definition yields a good model for depicting brine movement.

In Fig. 1, we illustrate the network definition by showing four two-dimensional slices from a representative sample. In all slices, black represents the air phase, white represents the brine phase, and gray represents the ice phase. Two example ellipses shown in red illustrate the definition of nodes with their corresponding radii. Black dashed lines depict the edge connecting nodes $p_{i1}$ and $p_{i2}$ and a few other representative edges connecting adjacent nodes. For each step downwards from a given node, the branch of the brine channel may grow or shrink, split into multiple branches, join with other branches, remain constant, or die, all with probabilities dependent upon the given node throat size, depth/temperature, and proximity to other nodes.

For a fixed pocket size $r$, we can compute probability distributions from this collection of data. For example, to calculate the probability that pockets of this size remain we simply divide the number of connections $(x_i^*, y_i^*, z_i^*, r) \rightarrow (x_i^*, y_i^*, z_i^* - 1, r)$ by the total number of brine pockets of size $r$ in the sample:

$$P_{remain}(r) = \frac{\#(x_i^*, y_i^*, z_i^*, r) \rightarrow (x_i^*, y_i^*, z_i^* - 1, r)}{\#\text{pockets of size r}}$$

Similarly, we can compute the probability of deaths, splits, and joins. Taken together, these form an annotated directed network with brine pockets as nodes and directed edges indicating the possible next steps, weighted by their probabilities. This network represents a Markov chain that statistically describes the ice core sample.

## 3   Results: $\mu$CT 3-D Imaging

Fig. 2 shows the object volume, definition, and shape of the brine phase for the Butter Point and Iceberg Site ice cores. The trends in the top half of the core are similar to what we expect since the temperature in the top half of the core is relatively





cold and the expected brine volume fraction is small. However, at around $100 - 120$ cm the brine volume fraction begins to increase and the expected c-shape profile begins to appear. Although this trend persists for a few samples, it does not continue as we would expect into the bottom of the core for the warmest temperature samples. This suggests that perhaps the Peltier cooling stage was not sufficiently warming the temperatures of those samples above approximately $-7$ °C. Since the average

temperature of the cold room housing the $\mu$CT scanner was $-8$ °C, either the cooling stage warming mode was not functional or was overcome by this ambient cooling. This may highlight that the cooling stage is not sufficiently warming the ice, but instead produces a slush that has x-ray attenuating properties between ice and brine. Segmenting all the slush with the brine phase (assuming it is possible to isolate only the slush from signal noise) leads to an overestimate of the brine phase and an inaccurate depiction of brine channel size and connectivity. Conversely, segmenting the slush with the ice phase leads to an

underestimate of the brine phase and also an inaccurate depiction of the brine channels. We used segmentation thresholds that split the difference as best as possible, recognizing that there was indeed error in segmentation for these warmer samples. Thus, we will treat data points at depths below roughly 120 cm with caution. Unfortunately, this encompasses the region where the brine volume fraction crosses the $5\%$ critical threshold, limiting our ability to examine the "Rule of Fives" in this manuscript.

Since the cooling stage did not significantly warm samples beyond $-7$ °C, we were not surprised that general trends shown

in Fig. 2 for all metrics did not differ significantly from the same samples scanned isothermally and presented in Lieb-Lappen et al. (2017) as the percolation threshold was not crossed. We observed that the brine phase specific surface area increased with depth, structure model index (a shape index quantifying similarity of objects to plates, rods, or spheres) was roughly 3 (indicative of cylindrical objects), structure thickness decreased, structure separation increased, and fractal dimension was roughly 2. The degree of anisotropy for brine channels is presented in Fig. 3. As expected, the brine phase had increased

anisotropy throughout the middle of the core. The measure of connectivity and Euler number of the brine phase given in Fig. 3 fluctuated greatly, particularly for the sample at 170 cm in the Iceberg Site core. We present a more detailed analysis of the connectivity for the brine channels in the following section using the mathematical brine network. From the metrics above, we conclude that brine channels are primarily cylindrical in shape with more branches at lower depths, consistent with previous observations (Lieb-Lappen et al., 2017).

We compared the $\mu$CT-measured brine volume fraction to expected values derived from the Frankenstein and Garner relationship relating temperature, salinity, and brine volume fraction (Frankenstein and Garner, 1967; Cox and Weeks, 1983). For this analysis, we used the core temperatures and salinity values measured in the field. This yielded expected brine volume fractions for the two cores shown in Fig. 4. From 0 cm to 120 cm, the measured brine volume fractions match the expected values remarkably well. However, below a depth of 120 cm in both cores, the expected brine volume fraction is greater than

that measured using $\mu$CT. For example, at a depth of 160 cm the expected brine volume fraction is 4.5 and 4.2 times greater than the values measured for samples from the Butter Point and Iceberg Site ice cores, respectively. This provides an estimate for the degree by which the cooling stage failed to heat the warmer samples in the $\mu$CT.





## 4 Results: Brine Network Model

### 4.1 Model Definitions

From the binarized images of the brine phase for the Butter Point and Iceberg Site ice cores, we created a mathematical network. We will use the term *network* to refer to the entire brine phase of a given sample and/or the entire brine phase of all
samples in a given core. For a given sample, we define each brine *channel* to be a single connected component in the brine network. The number of brine channels per sample ranged from 830 to 4800, with maximum numbers occurring in samples from the top and bottom of the cores. Previous work has shown that these brine channels often appear in layers or sheets spaced approximately $0.5 - 1.0$ mm apart due to the ice growth mechanism and original skeletal structure (Weeks and Ackley, 1982). A single brine channel is a complex web containing many different parts, which we will call the *branches* of the brine channel.
We will define a *join point* to be the node where two branches come together and a *split point* to be the node where a single branch splits into multiple branches. We note that flipping the perspective of movement from downwards to upwards changes a *split point* into a *join point* and vice versa. This is an important observation since we did not record the vertical orientation of the samples during cutting. Using this terminology, we use techniques from network theory to topologically characterize the brine network, gaining further insight into the complex connectivity and implications for permeability.
In the analysis below, we will examine several metrics that describe the topology of the brine network and are important for fluid flow implications. For example, we look at the throat sizes of the brine channels to gain an insight into the quantity of fluid that can move through different regions of a given brine channel. We then look at specific branches, both in terms of the number of branches and the size of each branch to learn more about the specific pathway available for fluid movement. As part of this analysis, we will investigate whether the likelihood of a given branch to split into multiple branches is dependent
upon the throat size of the parent branch. Finally, we will examine the size distribution of particular paths, looking for "pinch points" that may restrict flow and large regions that can provide maximize flow through the network. Together, these metrics will provide a detailed description of the micro-scale complexity of the brine network.

### 4.2 Throat sizes of channels and branches

For each brine channel we calculated the average throat size $\{\overline{r}_{z_i}\}$ for all nodes $\{p_{z_i}\}$ at a given depth in the sample. Fig. 5
and Fig. 6 show the collection of these throat sizes for the five largest (by vertical extent) brine channels at each depth for the Butter Point and Iceberg Site cores, respectively. There were 6 brine channels in each core that connected from the top to the bottom of the sample, with the majority of these channels found in samples from the top of each core. Generally speaking, the lengths of the vertical extent of the longest channel decreased with depth from the top of each core to around 60 cm, increased from 60 cm to roughly 120 cm, and then decreased from 120 cm to the bottom of the core. The trend between the top of the
core and roughly 120 cm is consistent with what we would expect due to brine volume fraction, temperature, and the expected c-shape profile. The channels from the lower depths, which had even warmer temperatures, did not reach in-situ temperatures during scanning as described previously. We also note that both the lengths of the vertical extent and average throat size quickly




diminished for channels beyond the few largest ones, as can be seen in the bottom panel of Fig. 5 and Fig. 6. Thus, we learn that fluid flow is most controlled by the behavior of the largest brine channel for a given section of sea ice.

The number of branches for a particular brine channel has potentially significant implications for fluid flow and permeability, such as influencing the rate at which chemical species may pass through the sea ice (Santiago et al., 2014; Yang et al., 1995; Newman, 2011). By increasing the number of branches, split points increase the number of potential paths through the sample. A higher number of paths increases the probability of finding a path connecting the top and bottom of a sample, thereby crossing the percolation threshold (Sahimi, 2011). Alternatively, split points can represent bottle-necks if the resulting child branches have smaller throat sizes than the parent throat, measured either as a minimum or as an aggregate. In each two-dimensional horizontal slice, we defined a node for each two-dimensional connected component. Each two-dimensional connected component corresponds to a distinct branch of the brine channel, and thus, the number of branches at a given depth (i.e. particular horizontal slice) equals the number of nodes at this depth. Fig. 7 shows the total number of nodes per depth for the 15 largest (by vertical extent) brine channels in the 70-cm sample of the Butter Point core. The largest channel had by far the largest number of branches, with a maximum of 20 branches at a depth of 2.1 mm from the bottom of the sample. Since this sample was from the columnar ice region, the maximum number of branches is relatively small. For comparison, the maximum number of branches for a sample in the frazil ice region at the top of the core was 124 nodes at a single depth. As expected, we find that brine channels in frazil ice have many more branches than brine channels in columnar ice, providing more distinct pathways for brine to move through the sample.

To gain insight into the behaviour of a channel, we visualized the number of branches and distribution of throat sizes by plotting the throat size $r_i$ of each node $p_i$ for the largest brine channel. Fig. 8 and Fig. 9 shows the throat sizes as a function of depth in the sample for three different representative sample depths: top, middle, and bottom of the Butter Point and Iceberg Site cores, respectively. For each channel shown, there is a plot of $\{r_i\}$ at each depth unsorted and a second corresponding plot sorted by descending $\{r_i\}$ for a given depth in sample. The first plots illustrate the connectivity of given branches, while the second plots provide a visualization of the distribution of $r_i$. The sample taken from the top of each core is from a region of frazil ice, which we would expect to have brine channels that are not well connected and have a distribution of throat sizes independent of depth in the sample. In both Fig. 8 and Fig. 9, panel a shows that the brine network for this top-most sample was indeed not well connected, while panel b shows that there was an even distribution of throat sizes. The two plots for mid-depth networks (70 cm) are quite similar, illustrating less tortuosity and easier ability to track particular branches in the brine channel. The bottom sample of the Iceberg Site core had much larger throat sizes, although this sample was an anomaly in Fig. 2 and Fig. 3. We did not observe a direct correlation between the number of branches and the throat size of those branches, as the distribution of throat sizes appeared to be more dependent upon the particular depth of the sample in the ice core, and consequently, the ice type of that sample. From this analysis, we learn that although there may be more branches for a given brine channel in frazil ice, the branches have better vertical connectivity in columnar ice. This means that fluid can more readily move upwards or downwards through the larger well-connected brine channels in columnar ice.



### 4.3 Probability distribution of branching nodes

Next we examined the branching of particular nodes to understand the behavior of particular fluid flow paths. Following a branch of a channel downwards, at an individual node the branch may end, continue onwards, or split into multiple branches. Conversely, by looking upwards, a node can be considered to be the first in a new branch, the continuation of a branch, or the joining point of multiple branches. Thus, for each node in a brine channel, we can tally the number of edges above and below said node to determine the degree of splitting or joining of branches in the channel (Newman, 2011). The large majority of nodes do not display branching, and the number of two-way splits/joins was roughly the same as the number of times a branch started/ended. We observed decay for frequency with increasing quantity of splits/joins. The Iceberg Site core had a larger number of higher order branching with a significant number of 7-way or 8-way splits/joins. A branch that splits is most likely to split into only two child branches, and thus for example, a contaminant introduced at a point source is likely restricted to a small horizontal region, following only a few separate paths through the ice. When a split occurred, we compared $r_i$ for the parent node to the collection of $r_i$ for the children nodes with similar behavior observed in both cores. $84\%$ of the time for the Butter Point core and $86\%$ of the time for the Iceberg Site, the sum of the throat sizes for the children node were greater than that of the parent node. However, the parent node was still larger than the largest child node $67\%$ of the time for the Butter Point core and $68\%$ of the time for the Iceberg site core. Thus, we learn that larger brine channels are more likely to split than smaller channels, and after the split, the fluid can access a larger region of the sea ice.

With knowledge of the total number of split points and join points, we then investigated the likelihood that branching was dependent upon the throat size. Fig. 10 shows the probability distributions for pockets dying, remaining, joining, and splitting for two regions of each of four samples, frazil ice and columnar ice. Note that for these plots, we used two additional first-year sea ice cores from previous work in addition to the Butter Point and Iceberg Site cores (Lieb-Lappen et al., 2017). The most basic difference between the two regions is that larger pocket sizes do not appear in columnar ice. However, for ranges of $r$ occurring in both regions, the shapes of the plots are similar. While small pockets can disappear, larger ones generally do not – the probabilities tend to zero (as marked by arrows in the plots in the first column). For pockets that remain but do not split or join with others, smaller pockets remain with lower probability (because more of them vanish) but then the probabilities follow an inverted parabolic trajectory, peaking around $r = 130 \ \mu m$. An interesting difference appears as throat size grows. In columnar ice, for throat size around $r = 500 \ \mu m$, we see two distinct behaviors. Some sizes remain with probability one (indicated by the top arrow in the second plot of the second row), while others remain with probability zero (bottom arrow). For the latter, looking at the last plot in the bottom row, we see these pockets are splitting into two or more pockets (indicated by the top arrow in that plot). This gives a signature for columnar ice – most brine channels simply continue on with slightly varying throat size but the ones that change generally split, creating a fork in the channel.

In frazil ice, the story for pockets which vanish is the same, as throat sizes become larger, they do not vanish in the next level. For remaining, splitting, and joining, however, there are new wrinkles in frazil ice relative to columnar ice. For pockets that remain, for larger throat sizes we see three types of behavior, two of which are similar to the behaviors in columnar ice (indicated by the top and bottom arrows of the second plot in the first row). But, a third behavior, where fifty percent of pockets



remain, is new for frazil ice (middle arrow). This new behavior is echoed in the probabilities of splitting and joining (indicated by the middle arrows in those plots) which shows that in this regime, brine channels have a complex behavior, remaining, splitting, and joining with high frequency. This third category of behavior for large throat sizes is a signature of frazil ice.

In addition, we summed the total number of edges leaving (splits) and entering (joins) each node over all nodes for the five largest brine channels of each sample. Fig. 11 plots these raw counts and the difference between the two are given for the Butter Point and Iceberg Site cores. When we consider split points and join points separately, we are considering the network as a graph with directed edges. The difference between the number of splits and the number of joins (i.e. difference between number of incoming and outgoing edges) is a metric for the topological complexity of a network (Newman, 2011). The raw counts for number of splits and joins both had roughly a c-shape profiles for both cores, with largest values and variability observed towards the bottom of the core. This is to be expected because the warmer part of the core allows for greater interconnectivity of branches in the brine network. For all brine channels, the number of splits was quite similar to the number of joins, and hence the differences between the two were quite small. However, there was still a general c-shape profile between 0 cm and 140 cm, indicating that topological complexity is greatest near the top and bottom of the core. This is consistent with frazil ice in the top of the core and increased branching in the warmer ice. Interestingly, both cores showed a decrease in topological complexity for the lowest samples below 140 cm. This could either be an artifact of not achieving actual in-situ temperatures with the cooling stage, or potentially an indication of a thought-provoking trend. If samples were not reaching in-situ temperatures, isolated channels may not have rejoined upon warming from storage temperatures, thereby reducing the number of split points and join points. Alternatively, a possible explanation of a real trend could be that as brine channels widen for the warmest samples, branches join together, reducing the topological complexity. A consequence of reducing the number of branches is a reduction in the number of split points and join points.

## 4.4 Capacity for fluid flow

We next examined the fluid flow capacity of each channel by both summing the number of pixels associated with all nodes for each channel and summing the total throat sizes of all nodes in each channel. We note that this represents a region larger than the pathways used for current fluid flow since many branches do not connect the top of a sample to the bottom. However, when the ice begins to warm and the branches become more interconnected, the process will likely start from the existing regions containing brine. Thus, this metric offers a starting place for comparing the capacity for fluid flow across different samples. Fig. 12 shows cumulative distribution functions for the number of brine channels as functions of the total number of pixels in the channel, with each line represents a different sample depth. The lines are colored on a gradient from red representing the top of the core to blue for the bottom of the core. The distribution functions for all depths on both cores were remarkably similar, and pairwise Kolmogorov-Smirnov tests did not detect that any two curves were from different probability distributions ($p \geq 0.1$) (Massey, 1951; Graham and Hogg, 1978). Both cores did show a trend of increased probability of brine channels with more pixels occurring at shallower depths, with a more robust trend observed in the Iceberg Site core. This trend could be due to samples at lower depths having an increased number of isolated small channels that have yet to connect to larger channels. Since there is doubt as to whether the samples below 120 cm were scanned at their in-situ temperatures, perhaps





these small isolated channels would have connected to larger channels under warmer conditions. Fig. 13 presents similar cumulative distribution functions for the number of brine channels as functions of the summed throat size of all nodes in the channel. The curves yield the same observations as before, with the Iceberg Site core again having a stronger correlation of increased probability of larger channels occurring at shallower depths. Likewise, Kolmogorov-Smirnov tests did not detect any two curves representing different probability distributions ($p \geq 0.1$). Any noticeable changes to the relative shape of the curves represent disproportionate changes in the shape of the brine channel with size of the channel, however, these variations were quite minor. In general, the shape of the curves in Fig. 13 are similar to those in Fig. 12. Thus, we conclude that brine channels in samples near the top of the core provide fluid with multiple distinct pathways to move through the sample, while deeper in the core there are only a few large channels with many small isolated paths that may connect under warmer ice conditions.

## 4.5 Following individual fluid flow paths

To further assess fluid flow capabilities, we analyzed individual branches of brine channels to isolate particular paths through the network. By construction, moving from $p_i$ to $p_j$ along an edge must either increase or decrease the height in the sample by one step (15 $\mu$m). First, we treated the network as a directed graph and considered paths starting from the first node. Since the network does not allow lateral movement, each step along an edge corresponds to a 15 $\mu$m step downwards. Although previously 6 brine channels were found that connected the top to the bottom of a given sample, no such paths were found in the directed graphs. This is because all paths connecting the top to the bottom of a sample required some movement upwards along a branch in order to reach the bottom. Fig. 14 shows an example of a brine channel where although the network is connected, any connecting path involves both upward and downward flow, such as the path highlighted in red. Thus, we selected the longest downward directed path from each brine channel, as well as any additional paths of the same length. This mimics a natural process such as gravity drainage, allowing us to study its influence on brine movement in the absence of pressure forces that aid upwards transport. Summing over all brine channels in the Butter Point core resulted in 63 763 directed paths. From this collection, we could selected the 15 316 paths of length 50 steps (750 $\mu$m) for statistical analysis of minimum throat size ($r^{min}$), maximum throat size ($r^{max}$), and summed throat size. Future work will enable us to use this model to statistically recreate brine channels that have this same statistical distribution of brine channel sizes.

We completed a similar analysis on the brine channel network, however this time treating it as an undirected graph with bidirectional edges. To avoid complexities arising from cycles (repeating loops), we only considered different spanning trees (paths that reached every node but have no cycles). We used a depth-first search algorithm to find all paths reaching the maximum vertical extent of each channel (West, 2001; Newman, 2011). We checked results through comparison of the distance obtained using Dijkstra's algorithm for finding the shortest-path tree (West, 2001; Dijkstra, 1959). This resulted in 36 449 paths over all the brine channels in the Butter Point core, of which 1753 were of length 50 steps (750 $\mu$m). We note that we can use the adjacency matrix to calculate the number of different walks (paths including cycles) that connected the top and bottom of a sample (West, 2001). However, due to the size of the adjacency matrix, this became computationally too expensive for large brine channels.



We used the 1753 paths of length $> 50$ steps to develop probability distributions for basic network statistics important for fluid flow such as $r^{min}$, $r^{max}$, and summed throat size of the path. These statistics can yield valuable information regarding the location and distribution of "pinch points," large channels, and total fluid flow through a brine channel. First, the paths were split into 3 categories based upon the ending node ($p_f$) throat size: $r_f < 1500$ $\mu$m, $1500 \leq r_f < 5250$ $\mu$m, and $r_f \geq 52\,500$

$\mu$m. Then, we split each category into 3 sub-categories based upon the starting node ($p_1$) throat size, using bins of the same size ($r_1 < 1500$ $\mu$m, $1500 \leq r_1 < 5250$ $\mu$m, and $r_1 \geq 52\,500$ $\mu$m). This division resulted in splitting the paths into 9 separate sub-categories. For each sub-category, we calculated the probability distributions for $r^{min}$, $r^{max}$, and summed throat size of the path and we present these histograms in Fig. 15. All plots are color-coded by $r_1$, with red, blue, and green histograms representing small, medium, and large throats, respectively. All histograms show the similar trend that both $r_1$ and $r_f$ have a

strong influence on resulting metrics, with larger $r_1$ and/or $r_f$ having larger $r^{min}$, $r^{max}$, and total volumes. $r_f$ had slightly more of an impact, particularly on $r^{min}$. There were no clear general trends in regards to the shape of the distributions. However, we note that for the smallest $r_f$, all three histograms for $r^{min}$ had a large peak around 30 $\mu$m (Fig. 15, top row. This peak corresponds to the smallest measurable branches, and we could potentially remove these paths from current fluid flow analysis. However, as the ice begins to warm, these "pinch points" are likely to have a significant impact on crossing

percolation thresholds.

## 5    Conclusions

The primary objective of this manuscript was to characterize the brine channel topology, morphology, and connectivity, thereby providing the statistical framework that we can use to create a sea ice brine channel network model. Since there was roughly a linear correlation between sample temperature and depth in the ice core, trends observed with one variable could easily

be converted to the other. Here we have presented quantitative metrics of the brine channel topological complexity, degree of connectivity, average and individual throat sizes, probability distributions for branches to split and join, capacity for fluid flow. The probability distributions shown represent a sampling of the various possibilities needed to statistically create a brine channel network parametrized by depth and temperature.

Overall, we observed similar profiles for both first-year sea ice cores. Both cores had the expected c-shape profile for throat

sizes. Toplogical complexity also had a similar c-shape profile that is consistent with complex frazil ice in the top of the core, relatively cold columnar ice below it, and increasingly warmer columnar ice at lower depths. We did not have good success in imaging and thresholding platelet ice as we were not able to reach the warmest in-situ temperatures at the bottom of the core. We observed more branching in frazil ice than columnar ice, but better vertical connectivity in the columnar ice and thus better capacity for fluid flow. By analyzing probability distributions for branching, we developed characteristic signatures for

each ice type. For columnar ice, most brine channels simply continue downwards with little change in throat size. However, for channels in which there is a significant change in throat size, there is a likely fork leading to a split in the channel. Meanwhile, large channels in frazil ice will split, join, and remain, all with relatively high frequencies. Columnar ice had fewer, but larger, channnels that showed increased connectivity with warmer ice at lower depths.



When examining the branching in brine channels, we observed that the largest channels had the greatest number of branches, but overall the throat size did not appear to have a direct correlation with the number of branches. Throat sizes were most dependent upon the depth and consequently ice type. When a split in a brine channel does occur, it is most likely to split into two child branches, and after the split, the brine generally has access to a larger region of the sea ice than before. Starting and ending throat sizes are strongly correlated with the flow capacity with larger initial/final throat sizes strongly indicative of increased flow. We detected pinch points in the brine channels that are critical points when determining whether the sea ice cover has crossed the percolation threshold. However, further work is needed in examining warmer ice with greater brine volume fractions.

The next step for this work is to create a brine channel network from the statistics presented here. For a sample at a given depth/temperature, first an initial set of nodes at the top height would be selected, where the nodes have throat sizes consistent with the probability distributions shown here. Branches could grow or shrink, split into multiple branches, join with other branches, remain constant, or stop, all with probabilities dependent upon the given node throat size, depth/temperature, and proximity to other nodes. The model described herein can help address questions such as how microstructural changes may be path dependent (e.g., whether to consider both upwards and downwards flow), how fluid flow may vary with depth, and what are the percolation implications of temperature fluctuations in an ice core. In summary, we successfully developed a method using $\mu$CT imaging to characterize the geometry of brine channels, whereby we can parameterize the pore networks using topological techniques that can be adjusted for depth and temperature, correlated with physical properties, and used in dynamical models of sea ice.

*Acknowledgements.* This research was supported by US National Science Foundation (NSF) grant PLR-1304134. The views and conclusions contained herein are those of the authors and should not be interpreted as representing official policies, either expressed or implied, of the NSF or the United States Government.





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





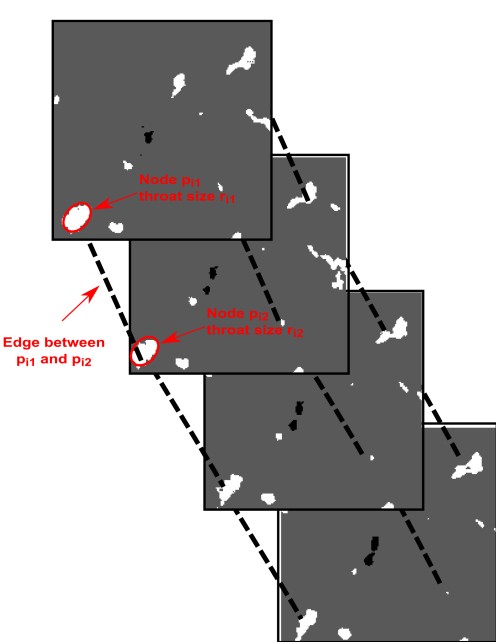

**Figure 1.** Sketch illustrating how the brine channel network is defined. Four horizontal two-dimensional slices are shown with lines connecting adjacent nodes (not all lines are drawn).





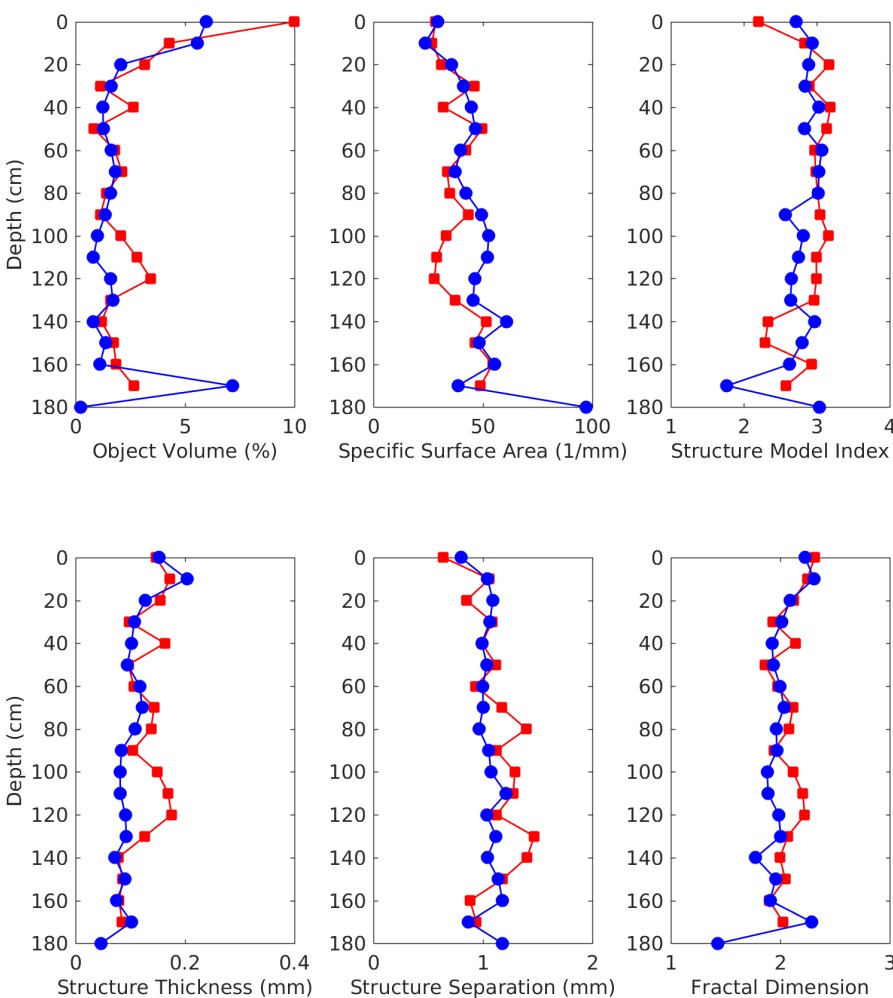

**Figure 2.** Definition and shape of brine channels scanned with a $\mu$CT scanner and a Peltier cooling stage set at in-situ temperatures. The red squares and blue circles represent ice cores from Butter Point and Iceberg Site, respectively.





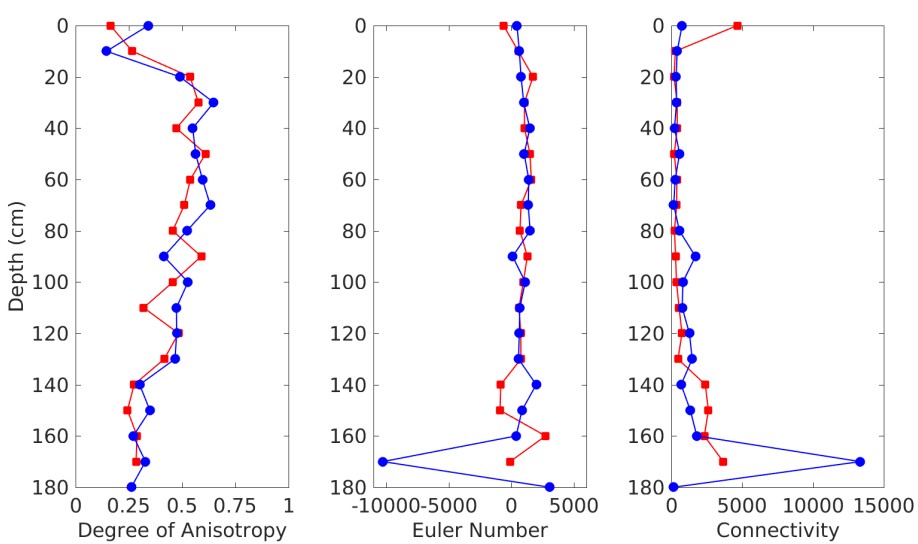

**Figure 3.** The degree of anisotropy, Euler number, and degree of connectivity of the brine channels for $\mu$CT scans at in-situ temperatures. The red squares and blue circles represent ice cores from Butter Point and Iceberg Site, respectively.



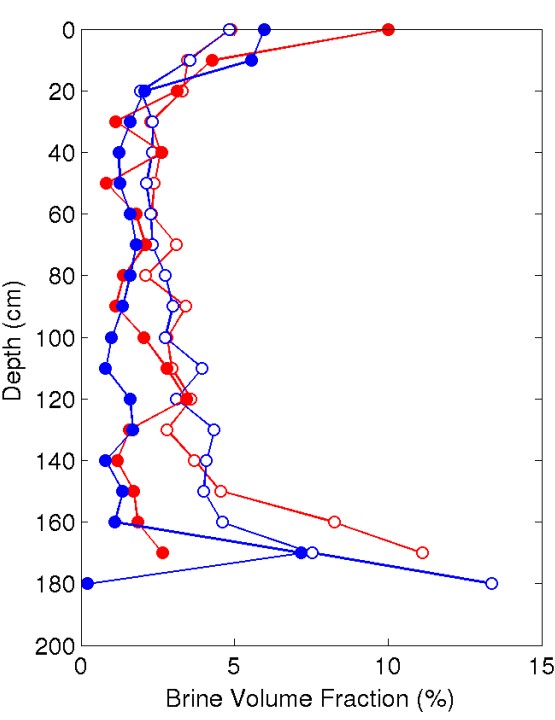

**Figure 4.** Comparing $\mu$CT-measured brine volume fraction to the expected values derived from the Frankenstein and Garner relationship (Frankenstein and Garner, 1967; Cox and Weeks, 1983). Results from the Butter Point (red) and Iceberg Site (blue) cores, where the $\mu$CT-measured values are the filled circles and the expected values are the open circles.





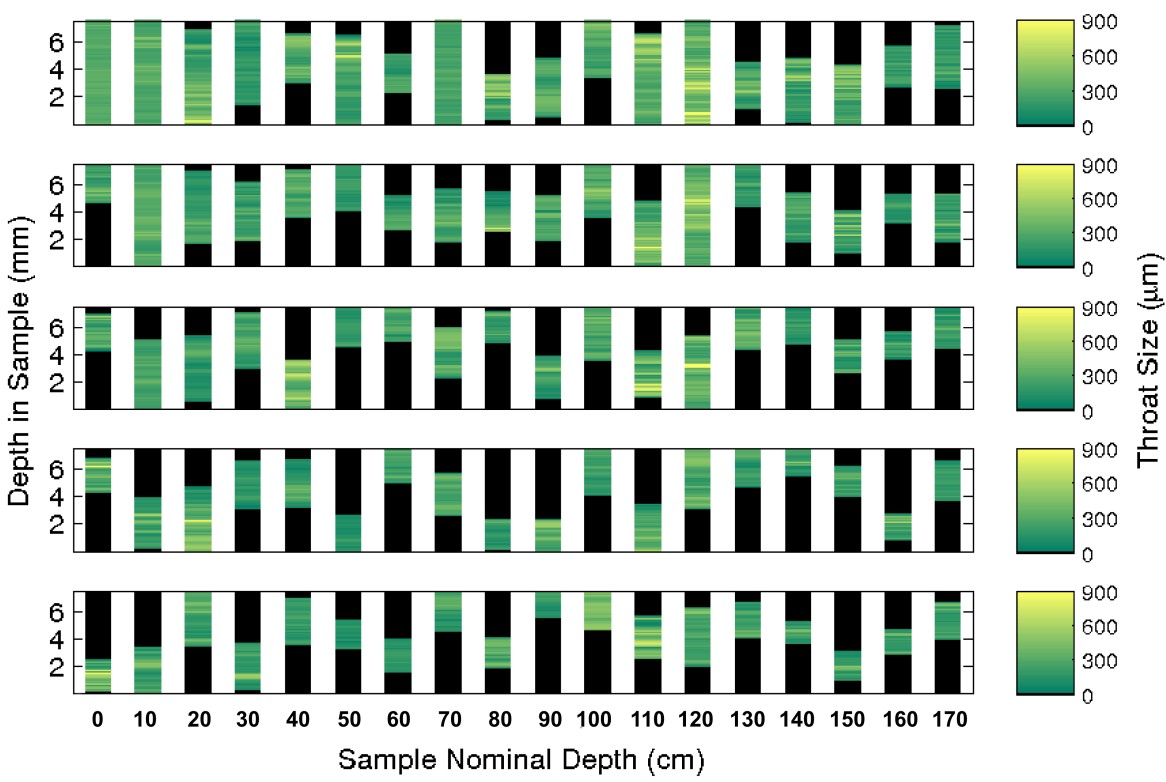

**Figure 5.** Average throat size $\{\bar{r}_{z_i}\}$ for the five largest brine channels of each sample for the Butter Point ice core. For each channel, at a given depth we calculated the average throat size of all nodes and color-coded accordingly. Note that there are only 6 channels that connect the top to the bottom of the sample (at depths of $0, 10, 70,$ and $120$ cm).



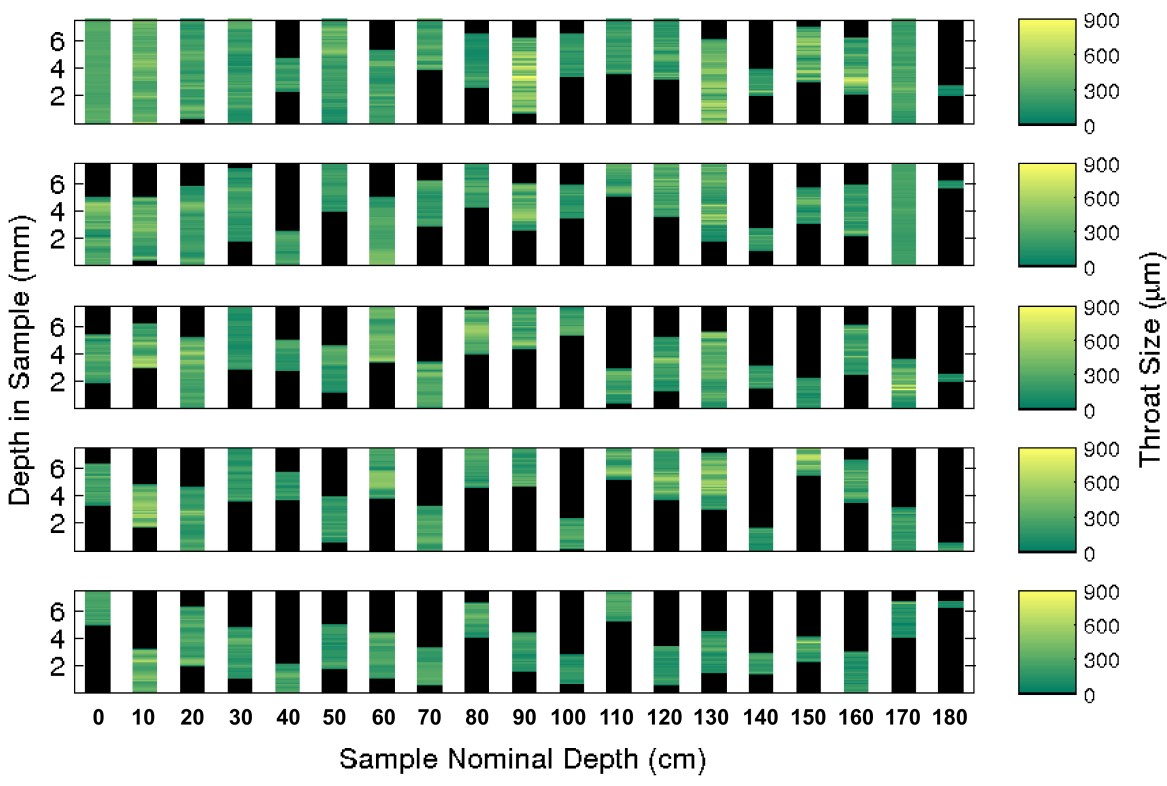

**Figure 6.** Average throat size $\{\overline{r}_{z_i}\}$ for the five largest brine channels of each sample for the Iceberg Site ice core. For each channel, at a given depth we calculated the average throat size of all nodes and color-coded accordingly. Note that there are only 6 channels that connect the top to the bottom of the sample (at depths of $0, 10, 30, 50,$ and $170$ cm).





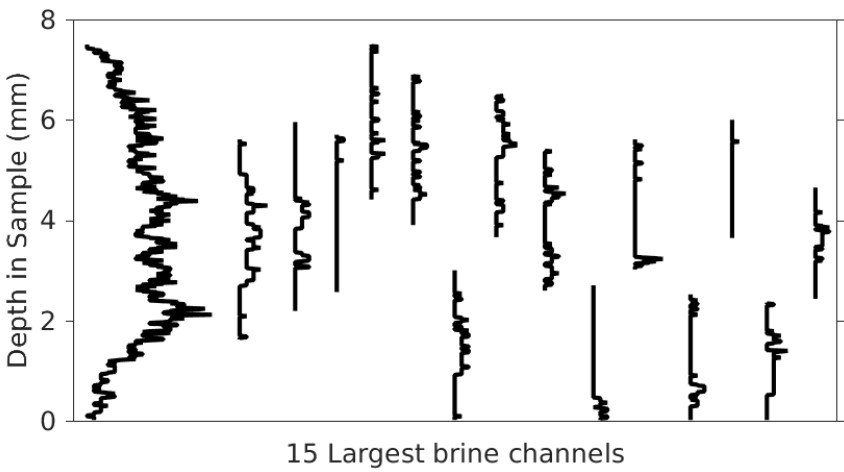

**Figure 7.** Number of branches at each depth for the 15 largest brine channels in the 70-cm sample of the Butter Point core. We measured the number of branches by the number of nodes at a given depth in the sample. Each line represents a separate channel, and the horizontal extent illustrates the number of branches.





**Figure 8.** Throat size $r_i$ of each node for the largest channel of representative samples in the Butter Point core. The top, middle, and bottom rows show the largest brine channel from the sample at $0, 70$, and $170$ cm, respectively. The left panels show the throat sizes at each depth in the sample unsorted, while the right panel sorts the throat sizes in descending order.





**Figure 9.** Throat size $r_i$ of each node for the largest channel of representative samples in the Iceberg Site core. The top, middle, and bottom rows show the largest brine channel from the sample at $0, 50,$ and $170$ cm, respectively. The left panels show the throat sizes at each depth in the sample unsorted, while the right panel sorts the throat sizes in descending order.





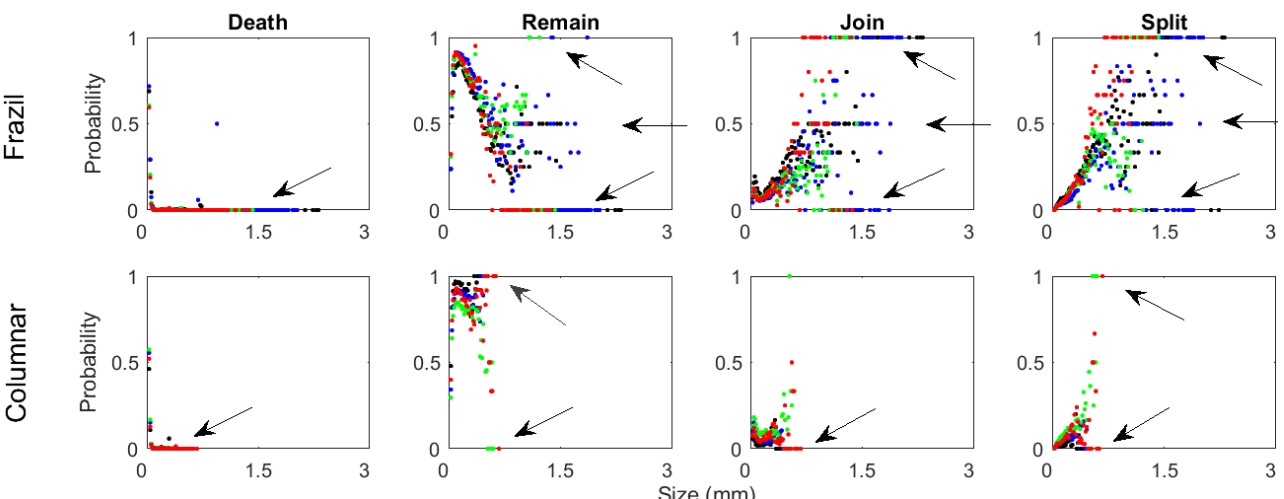

**Figure 10.** $\{r_i\}$ probability distributions showing the likelihoood a node dies, remains, splits, or joins as a function of the throat size $r$. The top four figures show probability distributions for frazil ice while the bottom four figures show probability distributions for columnar ice. Note that to complete these figures, in addition to the Butter Point and Iceberg Site cores, we used two additional first year sea ice cores from previous work (Lieb-Lappen et al., 2017). Butter Point is shown in black, Iceberg Site is shown in blue, and the two additional first year ice cores are shown in red and black.



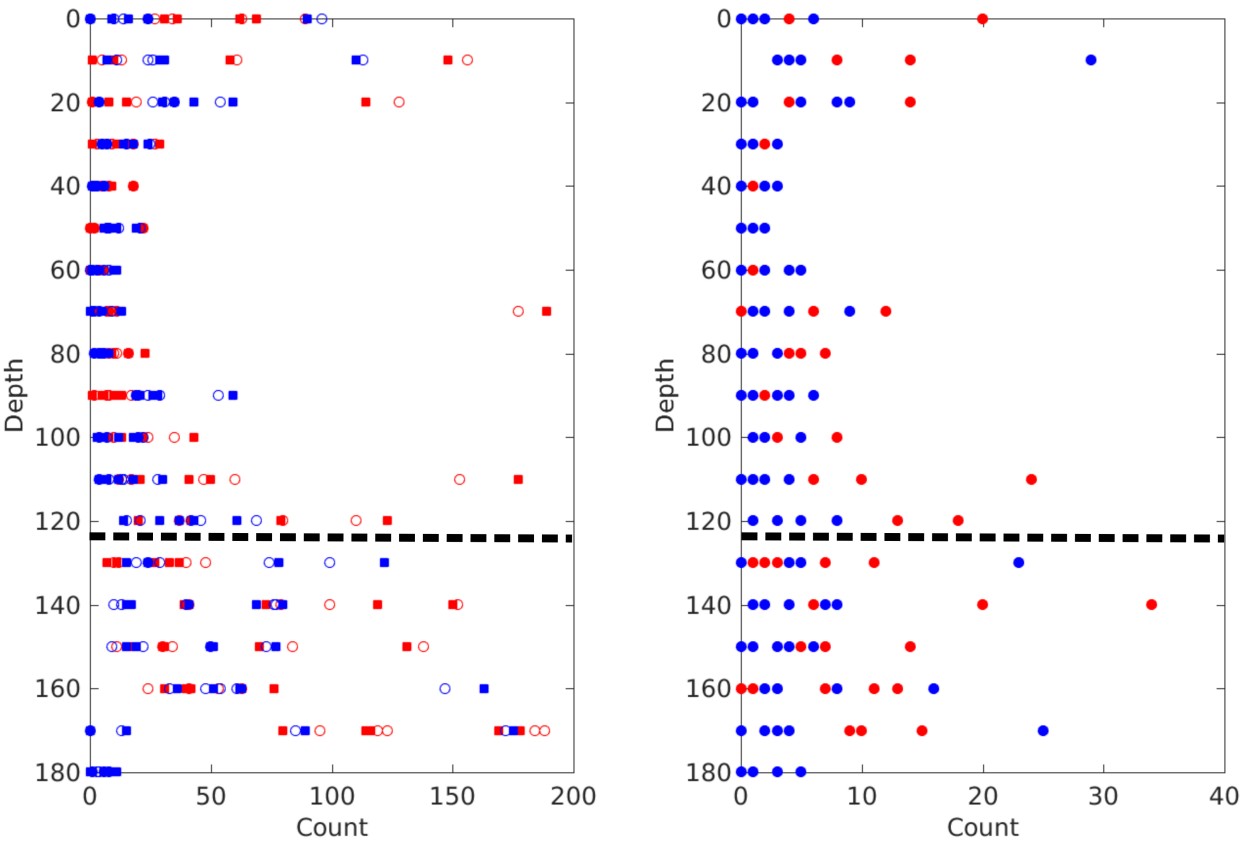

**Figure 11.** Topological complexity of the five largest brine channels in each sample for both the Butter Point (red) and Iceberg Site (blue) cores. The left panel shows the total number of splits (open circles) and joins (black squares) over all nodes in a given channel. The right panel shows the absolute value of the difference between the number of splits and joins. The dashed line highlights the depth below which there is concern regarding the effectiveness of the cooling stage and whether samples were scanned at actual in-situ temperatures.





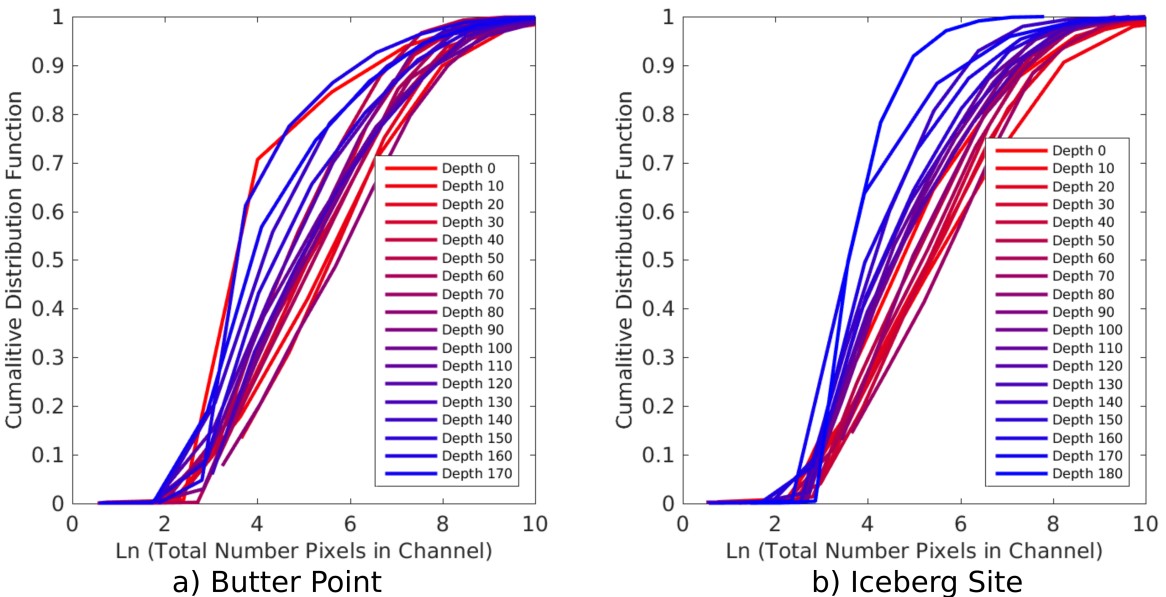

**Figure 12.** Cumulative distribution functions for number of brine channels as functions of the total number of pixels in the channel. The left and right panels are for the Butter Point and Iceberg Site ice cores, respectively. In both panels, each line represents a different sample depth where the lines are colored on a gradient from red representing the top of the core to blue for the bottom of the core. Note that pixels in the original $\mu$CT images are 15 $\mu$m on each edge.





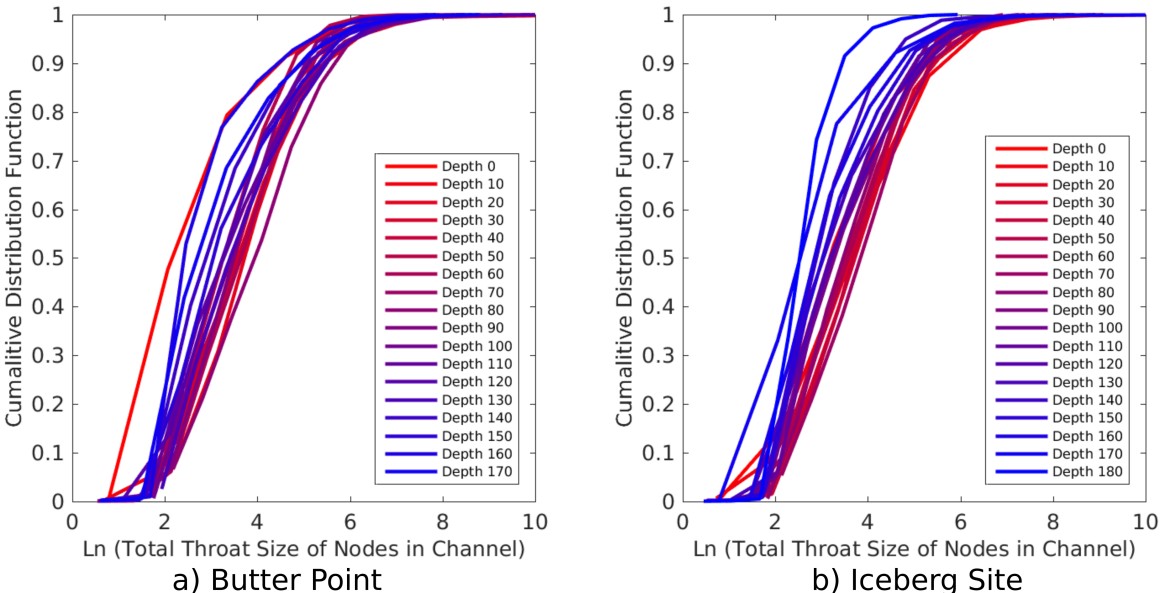

**Figure 13.** Cumulative distribution functions for number of brine channels as functions of the summed throat size of all nodes in the channel. The left and right panels are for the Butter Point and Iceberg Site ice cores, respectively. In both panels, each line represents a different sample depth where the lines are colored on a gradient from red representing the top of the core to blue for the bottom of the core. Note that throat sizes in the original $\mu$CT images are 15 $\mu$m on each edge.



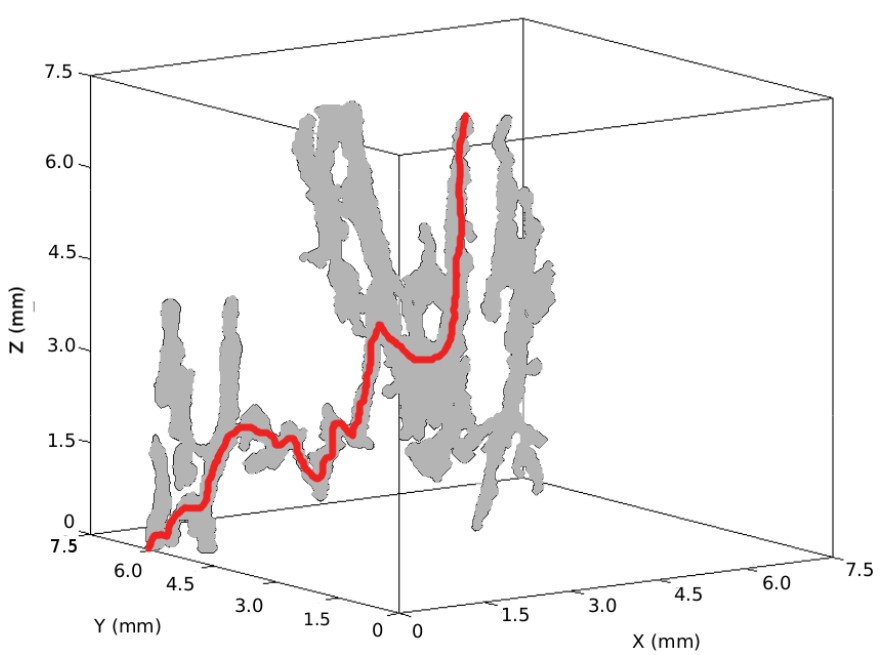

**Figure 14.** Largest brine channel at 70 cm in the Butter Point ice core. Although this brine channel connects from top to bottom, there is not

a directed path that does so. Any connecting path involves movements both upwards and downwards. One such path is highlighted in red.



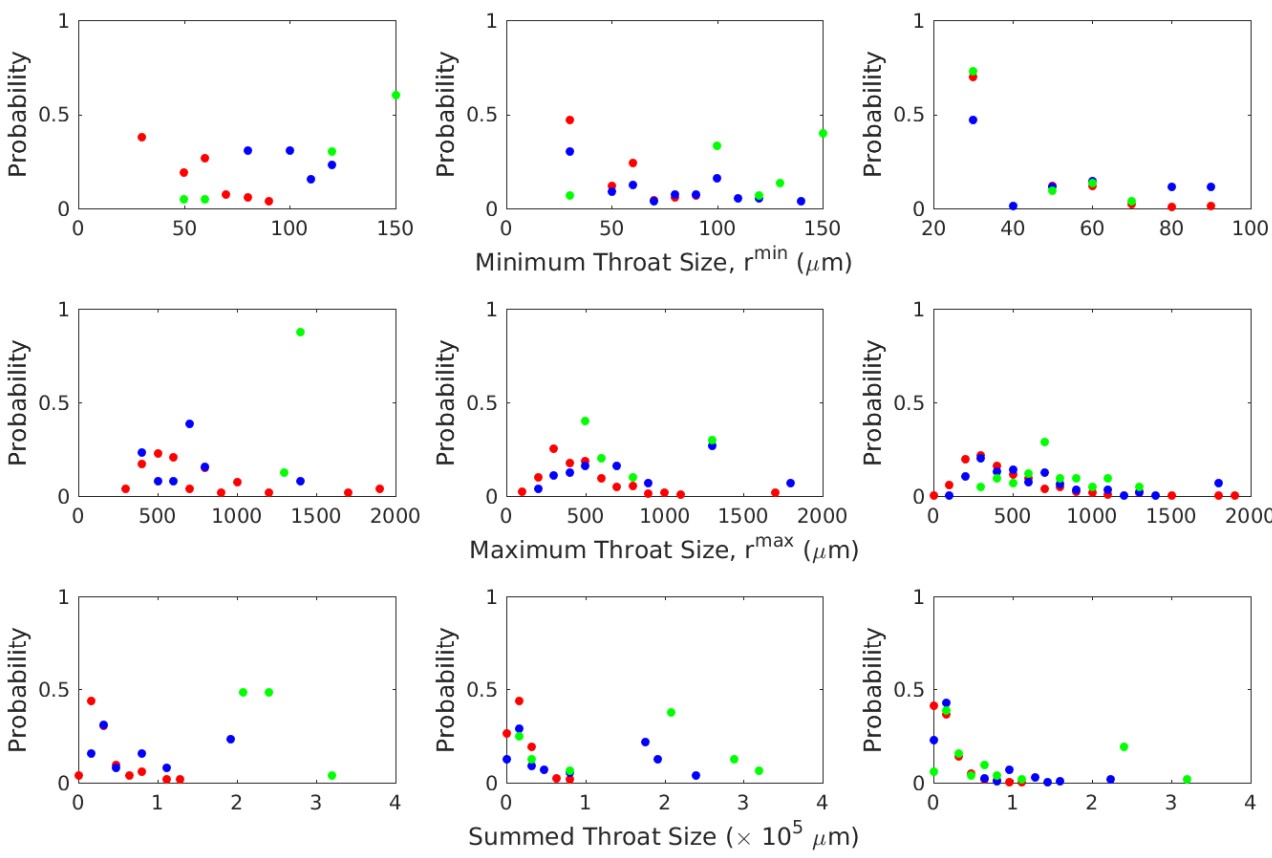

**Figure 15.** Probability distributions of paths connecting the top to the bottom for all brine channels in the Butter Point ice core. Only paths greater than 50 steps, or 750 $\mu$m were considered. Left, middle, and right panels represent channels where $r_f < 1500\ \mu$m, $1500 \leq r_f < 5250$ $\mu$m, and $r_f \geq 52\,500\ \mu$m, respectively. Top, middle, and bottom rows represent probability distributions for $r^{min}$, $r^{max}$, and summed throat size, respectively. For all panels, the colors red, blue, and green represent channels where $r_1 < 1500\ \mu$m, $1500 \leq r_1 < 5250\ \mu$m, and $r_1 \geq 52\,500\ \mu$m, respectively.