# Peer review of "A Network Model for Characterizing Brine Channels in Sea Ice"

_The Cryosphere, 2017_

## Referee Comment (RC1) · Anonymous Referee #1 · 23 Sep 2017

General Comments:

This paper presents a topological and statistical analysis of novel three dimensional images of sea ice microstructure. In this paper, a directed graph is mapped to the microstructure of the sea ice and a throat size assigned to each node from the semi-minor axis of a best fit ellipse around a brine pocket viewed from a horizontal slice. Edges are assigned when moving though the ice if a brine pocket continues immediately below the previous elevation. Edges themselves can represent a splitting or joining of a brine channel depending on the overlap of the pockets being compared. Using this network model, statistical analysis is carried out relating morphological characteristics to depth and temperature. The results of the analysis are consistent with observed characteristics of sea ice microstructure for both columnar and frazil ice. The analysis presented

here is then to be used as a basis for model development.

This paper presents an extensive analysis of rare and difficult to obtain microstructural data. Sea ice microstructure moderates a broad range of physical processes in both the Arctic and Antarctic. As a result, this statistical analysis should be of broad interest to polar science community. Overall, the analysis is well thought out, and well executed. I recommend this paper for publication with the following specific issues being addressed and or considered. I might also suggest the authors carefully read through the manuscript, there are sporadic minor grammatical errors.

Specific Comments:

Line 1: "The brine network in sea ice is a complex labyrinth whose..."

I understand what you are trying to get at here but the description is not completely accurate. A labyrinth would imply that there is no order to the channel development, this is not the case.

It might be better to say something like, " The brine pore space in sea ice can form complex connected structures whose geometry is critical in the governance of important physical transport processes between the ocean, sea ice and surface."

Page 2.

Line 32: "since viewed in two-dimensional slices"

I think it would be prudent to add the fact that these are horizontal slices for clarity.

Page 3 Line 4: "This definition captures both the location and the size of the brine phase at any point..."

I'm not sure I understand this. The brine phase refers to the whole of the brine pore space. I think you might mean that it captures the location and size of a brine pocket

at any point. However, this is a 2d slice so you may want to find some other type of phrasing.

Near Line 25: The probability of remaining notation is a bit awkward and phrasing confusing. Saying you are counting the number of connections made me think of the total number of connections that can trace back to that pocket.

How about: For example, to calculate the probability that brine pockets of a given size remain we simply divide the number of pockets for a fixed throat size r that connect once from $z_i^*$ to $z_i^* - 1$ by the total number of pockets of that size.

Markov chain: I agree that the network model is representing a Markov chain, and that seems to make sense, it should be that way physically I think. But is there a physical justification that this is true? I believe it would strengthen the paper.

Section 3 Results: You might want to add a reference with definitions of the some of the quantities listed in Figure 2.

Page 4 Line 7: " but instead produces a slush that has x-ray attenuating properties between ice and brine".

Is the slush in the pore space? The wording makes it sounds as if the whole thing is slush, that of course would not make sense at -7 C. It becomes clear later, but saying that the slush is in the pore space immediately would make things clearer.

Could it also be that at the bottom of the core you had more brine leakage at the time of extraction? In this case ,what was in the brine space may have been less saline and thus slushy at the in-situ temperatures.

Line 11: It would be helpful to describe what is meant by "as best as possible".

Line 27: "Salinity Values Measured in the Field"

Were these bulk salinity measurements from adjacent cores? That should be stated if so.

Line 13: "This is an important observation since we did not record the vertical orientation of the samples during cutting"

Can't you tell by the direction of splitting, or are the samples too small to see that structure?

Line 21: Change maximize to maximum.

Line 22: The description of Figures 8 and 9 should be rethought and made more clear. It is not clear to me what the unsorted figures represent. I assume node index is just a way to label each node and to me it seems arbitrary. Ordering them by size makes sense but what gives the unsorted part of the figure any relevance? I think a description of how each node is labeled in the unsorted figure is needed to understand what it is meant to represent. Is it done by physical distance from the node with the largest throat size? It was not clear to me. This may all be fixed by clearly defining "node index" which I did not see in the figure.

Line 22 "The probability distributions shown represent a sampling of the various possibilities. . ."

Consider rephrasing, maybe change possibilities to microstructural behaviors?

For your future model development, you might want to consider the effect salinity might have on the statistics you consider. It is encouraging to see that the other two previous cores you use do follow the most recent though. However, in the Arctic summer snow melt can get into the pore space decreasing salinity and reducing permeability. Just a thought.

Figures: Figure 1: A figure showing how a split or join is assigned would be nice but is not completely necessary.

Figure 2. Rename object volume to brine volume fraction?

Figure 14: Just a comment, the up and down motion might be able to be captured if you had someway to include horizontal edges in your network model.

Figure 15: Connecting the dots with thin lines may make the figure easier to read, not sure, it is ok as is though.

Technical Corrections

Line 25: produces should be produce. "Since different growth rates in natural sea ice produce…"

Line 4: Remove the word "sufficiently"

Line 6: change "this ambient cooling" to "the ambient temperature".

Page 5 Line 21: Change maximize to maximum.

Figures

Figure 10: X-axis title is cut off a bit.

Figure 11: In the caption "black squares" should be "blue squares".

---

## Author Comment (AC1) · 16 Oct 2017

**RESPONSES TO REFEREE 1**

**General Comments:**

This paper presents a topological and statistical analysis of novel three dimensional images of sea ice microstructure. In this paper, a directed graph is mapped to the microstructure of the sea ice and a throat size assigned to each node from the semi-minor axis of a best fit ellipse around a brine pocket viewed from a horizontal slice. Edges are assigned when moving though the ice if a brine pocket continues immediately below the previous elevation. Edges themselves can represent a splitting or joining of a brine channel depending on the overlap of the pockets being compared. Using this network model, statistical analysis is carried out relating morphological characteristics to depth and temperature. The results of the analysis are consistent with observed characteristics of sea ice microstructure for both columnar and frazil ice. The analysis presented here is then to be used as a basis for model development.

This paper presents an extensive analysis of rare and difficult to obtain microstructural data. Sea ice microstructure moderates a broad range of physical processes in both the Arctic and Antarctic. As a result, this statistical analysis should be of broad interest to polar science community. Overall, the analysis is well thought out, and well executed. I recommend this paper for publication with the following specific issues being addressed and or considered. I might also suggest the authors carefully read through the manuscript, there are sporadic minor grammatical errors.

We thank Referee #1 for the overall comments and thorough reveiw of the manuscript.

**Specific Comments:**

Page 1, Line 1: "The brine network in sea ice is a complex labyrinth whose. . ." I understand what you are trying to get at here but the description is not completely accurate. A labyrinth would imply that there is no order to the channel development, this is not the case. It might be better to say something like, "The brine pore

space in sea ice can form complex connected structures whose geometry is critical in the governance of important physical transport processes between the ocean, sea ice and surface."

We have made the recommended change to the first sentence of the abstract in the revised manuscript.

**Page 2, Line 32: "since viewed in two-dimensional slices" I think it would be prudent to add the fact that these are horizontal slices for clarity.**

We have added the word "horizontal" in front of slices in the revised manuscript.

**Page 3, Line 4: "This definition captures both the location and the size of the brine phase at any point. . ." Im not sure I understand this. The brine phase refers to the whole of the brine pore space. I think you might mean that it captures the location and size of a brine pocket at any point. However, this is a 2d slice so you may want to find some other type of phrasing.**

We thank Referee $^\#1$ for identifying this confusing terminology. We edited the last sentence of this paragraph and the sentence now reads as follows:

*This definition captures both the location and quantity of the brine at any point in the sea ice.*

**Page 3, Near Line 25: The probability of remaining notation is a bit awkward and phrasing confusing. Saying you are counting the number of connections made me think of the total number of connections that can trace back to that pocket. How about: For example, to calculate the probability that brine pockets of a given size remain we simply divide the number of pockets for a fixed throat size r that connect once from $z_i$ to $z_i 1$ by the total number of pockets of that size. Markov chain: I agree that the network model is representing a Markov chain, and that seems to make sense, it should be that way physically I think. But is there a physical justification that this is true? I believe it would strengthen the paper.**

We have edited the definition of calculating the probability of remaining in the revised manuscript as suggested by Referee #1. In regards to the Markov chain, we agree that it seems to make sense physically due to the downward growth mechanism of sea ice. However, a complete justification is beyond the scope of this paper.

**Section 3 Results: You might want to add a reference with definitions of the some of the quantities listed in Figure 2.**

We have added the sentence below to the start of Section 3, referencing definitions for all quantities listed in Figure 2.

*We first used standard morphological metrics as defined in previous work to describe the brine network shape and size (Lieb-Lappen et al., 2017)*

**Page 4, Line 7: "but instead produces a slush that has x-ray attenuating properties between ice and brine". Is the slush in the pore space? The wording makes it sounds as if the whole thing is slush, that of course would not make sense at -7 C. It becomes clear later, but saying that the slush is in the pore space immediately would make things clearer. Could it also be that at the bottom of the core you had more brine leakage at the time of extraction? In this case, what was in the brine space may have been less saline and thus slushy at the in-situ temperatures.**

We have added a a clarification that the slush is in the pore space. During extraction, we did not notice any brine leakage, but we also can not definitively state that there was none. It is possible that the brine space may have been less saline and thus slushy at the in-situ temperatures but we have no direct observations to support this statement.

**Page 4, Line 11: It would be helpful to describe what is meant by "as best as possible".**

We have edited the identified sentence to be more accurate, and the sentence reads as follows in the revised manuscript:

*We used segmentation thresholds that split the difference with a threshold halfway between the peak of the brine phase and the peak of the ice phase,*

*recognizing that there was indeed error in segmentation for these warmer samples.*

**Page 4, Line 27: "Salinity Values Measured in the Field" Were these bulk salinity measurements from adjacent cores? That should be stated if so.**

We thank Referee #1 for catching this error. Bulk salinity was estimated from ion chromatography measured chloride concentrations. The identified paragraph has been edited and now starts as follows:

*We compared the μCT-measured brine volume fraction to expected values derived from the Frankenstein and Garner relationship relating temperature, salinity, and brine volume fraction (Frankenstein and Garner, 1967; Cox and Weeks,1983). For this analysis, we used the core temperatures measured in the field and salinity values estimated from ion chromatography measured chloride concentrations presented in Lieb-Lappen and Obbard (2015).*

**Page 5, Line 13: "This is an important observation since we did not record the vertical orientation of the samples during cutting" Cant you tell by the direction of splitting, or are the samples too small to see that structure?**

The samples were indeed too small to see that structure. We know the orientation of the vertical $z$-axis. However, unfortunately we did not have the direction of the vertical $z-$axis.

**Page 5, Line 21: Change maximize to maximum.**

We have made the recommended change in the revised manuscript.

**Page 5, Line 22: The description of Figures 8 and 9 should be rethought and made more clear. It is not clear to me what the unsorted figures represent. I assume node index is just a way to label each node and to me it seems arbitrary. Ordering them by size makes sense but what gives the unsorted part of the figure any relevance? I think a description of how each node is labeled in the unsorted figure is needed to understand what it is meant to represent. Is it done by physical distance from the node with the**

**largest throat size? It was not clear to me. This may all be fixed by clearly defining "node index" which I did not see in the figure.**

We have edited the language to clearly define the sorting in Figures 8 and 9 in the revised manuscript. The following has been added to the captions for Figures 8 and 9:

*The left panels show the throat sizes at each depth in the sample with nodes sorted by location, not by size. The right panel sorts the nodes by throat size in descending order.*

Additionally, the third paragraph of section 4.2 now starts as follows:

*To gain insight into the behaviour of a channel, we visualized the number of branches and distribution of throat sizes by plotting the throat size $r_i$ of each node $p_i$ for the largest brine channel. Fig. 8 and Fig. 9 shows the throat sizes as a function of depth in the sample for three different representative sample depths: top, middle, and bottom of the Butter Point and Iceberg Site cores, respectively. For each channel shown, there is a plot of $\{r_i\}$ at each depth sorted by physical location in a two-dimensional grid (working line by line), not by size. A second corresponding plot shows node sizes sorted by descending $\{r_i\}$ for a given depth in the channel. The first plots illustrate the connectivity of given branches, while the second plots provide a visualization of the distribution of $r_i$.*

**Page 10, Line 22: "The probability distributions shown represent a sampling of the various possibilities. . ." Consider rephrasing, maybe change possibilities to microstructural behaviors? For your future model development, you might want to consider the effect salinity might have on the statistics you consider. It is encouraging to see that the other two previous cores you use do follow the most recent though. However, in the Arctic summer snow melt can get into the pore space decreasing salinity and reducing permeability. Just a thought.**

We have made the recommended change to the identified sentence. We thank Referee #1 for the observation in regards to future model development and will indeed strive to include salinity in future development.

**Figure 1: A figure showing how a split or join is assigned would be nice but is not completely necessary.**

We thank Referee #1 for this suggestion but we feel as though the written description of a split and a join was sufficient and did not require an additional figure.

**Figure 2: Rename object volume to brine volume fraction?**

We have made the suggested recommendation in the revised manuscript.

**Figure 14: Just a comment, the up and down motion might be able to be captured if you had someway to include horizontal edges in your network model.**

We thank Referee #1 for this observation. We are intrigued by a definition for a horizontal edge, but it is not possible with the model as currently defined.

**Figure 15: Connecting the dots with thin lines may make the figure easier to read, not sure, it is ok as is though.**

With so many values having zero probability or no data, we made the stylistic choice to not include thin lines between the dots.

**Technical Corrections:**

**Page 1, Line 25: produces should be produce. "Since different growth rates in natural sea ice produce. . ."**

We thank Referee #1 for catching this error and have made the recommended revision.

**Page 4, Line 4: Remove the word "sufficiently"**

We have removed the word "sufficiently" in the revised manuscript.

**Page 4, Line 6: change "this ambient cooling" to "the ambient temperature".**

We have made the recommended revision in the revised manuscript.

**Page 5, Line 21: Change maximize to maximum.**

We have made the recommended change in the revised manuscript.

**Figure 10: X-axis title is cut off a bit.**

We have fixed the X-axis in Figure 10 in the revised manuscript.

**Figure 11: In the caption "black squares" should be "blue squares".**

We thank Referee #1 for catching this oversight and have edited the caption to read "filled squares".

---

## Referee Comment (RC2) · Anonymous Referee #1 · 6 Nov 2017

I am pleased to recommend this paper for publication after reading the author's responses. I believe they have addressed all of my previous concerns.

---

## Editor Comment (EC1) · J. Hutchings (Editor) · 6 Nov 2017

It has taken longer than usual to receive two reviews of this paper. My apologies for the delay in response, and I want to thank reviewer 1 for their timely response and opinion on the paper after consideration of their comments.
* * *

---

## Short Comment (SC1) · 30 Nov 2017

This is potentially a useful paper based on producing a network model of sea ice brine microstructure. However, I have found it extremely difficult to review. My main criticism being that based on network theory as it is, and with many of the references relating to this (e.g. Newman, 2011; Pierret et al. 2002; Delerue et al. 2003), much of the terminology and techniques will be completely unfamiliar to the general reader of The Cryosphere. I believe therefore that before full publication the manuscript needs to be restructures in a form that will make it much less opaque to readers who are not familiar with the methods presented. Without this I feel that any impact that it might have will be substantially diminished. I give below a series of specific comments which are intended to indicate what I see are some of the major issues and, in part, how they

might be addressed.

The first three comments relate to what I believe should be a standard introduction to the sampling and presentation of the initial measurements.

1. Please describe the ice cores. In other words, in each core, what was the thickness of frazil ice at the top, the thickness of columnar ice beneath, and any indication of platelet ice at the base? This is important in terms of reference for the interpretation of the inferred microstructure.

2. Were the cores sampled on site or after transportation? If the latter then please give details of how the cores were treated between extraction and sampling.

3. In section 3, with reference to Figures 2 and 3, please explain what the different parameters plotted are, what you might expect them to show, and how they relate to each other. For example, what is expected to be the relationship between brine volume fraction and specific surface area (Spor). In investigating permeability in sandstone samples Zhang and Weller (2014)* have demonstrated that there is a relationship between fractal dimension and Spor. Would any such relationship be expected here? Explain the Euler number for those unfamiliar with it. How is the degree of anisotropy derived? Given that the lower parts of the cores are explained to have a significant degree of uncertainty associated with the measurements (quantify this?) does it make sense for the scales for Euler number and connectivity to be dictated by the lowermost sampes?

*Zhang & Weller, 2014. Geophysics, 79, D377-D387.

The above comments are in fact relatively introductory and indicate the need for a clear explanation of the background to the study, how the initial sampling was carried out, and what the initial measurements show. Unfortunately I find that from this point onwards the manuscript becomes confusing and is not at all well explained or illustrated for the more general reader.
4. I find section 4.2 extremely confusing, including the figures that accompany it. Figures 5 and 6 ostensibly show the 5 "largest" brine channels from each of the two cores. I assume that one row represents 1 brine channel shown as a sequence of samples through the core in depth order. However, the colour scale suggests that the throat size goes to zero at many points i.e. a pore terminates. I find this hard to understand in the context of the fact that a single row represents a "large" brine channel. Please clarify.

The discussion on branches, with reference to Figure 7, is similarly very confusing. In Figure 7 what is the horizontal scale for each separate line representing a brine channel? What is the significance of the fact that one brine channel appears to pass almost right through a sample but the others do not?

I cannot at all understand the significance of Figures 8 and 9.

5. Similar comments apply to sections 4.3, 4.4 and 4.5, and their associated Figures. An excellent example of network terminology that will be unfamiliar to most is the statement "...treated the network as a directed graph..." etc.

In essence therefore, although I believe that the work presented is ultimately publishable, to be so requires considerable restructuring of the manuscript and I urge the authors to do this. There needs to be a much clearer explanation of the techniques, probably a reduction in the number of figures including clear explanations of what they represent. A somewhat broader review of previous work on looking at the interconnectness of brine channels in sea ice would also not be remiss.

---

## Author Comment (AC2) · 30 Dec 2017

**RESPONSES TO REFEREE 2**

**General Comments:**

**This is potentially a useful paper based on producing a network model of sea ice brine microstructure. However, I have found it extremely difficult to review. My main criticism being that based on network theory as it is, and with many of the references relating to this (e.g. Newman, 2011; Pierret et al. 2002; Delerue et al. 2003), much of the terminology and techniques will be completely unfamiliar to the general reader of The Cryosphere. I believe therefore that before full publication the manuscript needs to be restructures in a form that will make it much less opaque to readers who are not familiar with the methods presented. Without this I feel that any impact that it might have will be substantially diminished. I give below a series of specific comments which are intended to indicate what I see are some of the major issues and, in part, how they might be addressed. The first three comments relate to what I believe should be a standard introduction to the sampling and presentation of the initial measurements.**

We thank Referee #2 for the overall comments and thorough reveiw of the manuscript. We have significantly rewritten the methods section in the revised manuscript to make it more accessible to a wider readership.

**Specific Comments:**

**1. Please describe the ice cores. In other words, in each core, what was the thickness of frazil ice at the top, the thickness of columnar ice beneath, and any indication of platelet ice at the base? This is important in terms of reference for the interpretation of the inferred microstructure.**

Full descriptions of the ice cores including frazil, columnar, and platelet fractions were reported in previous papers (Obbard et al., 2016 and Lieb-Lappen et al., 2017). We have added two sentences to the first paragraph of the methods section stating these fractions and referencing the relevant papers. The first paragraph of the methods section now starts as follows:

*This work will focus on two of the ice cores extracted from different locations in the Ross Sea, Antarctica during a October - November* 2012 *field campaign. The* 1.78 *m Butter Point ice core was collected at* 77°35.133′ *S and* 164°48.222′ *E and had a temperature gradient ranging from* −16.1 °*C at the top to* −2.5 °*C at the bottom. For this core, the top* 14 *cm was frazil ice, the columnar ice region was from* 14 *cm to* 65 *cm, and platelet ice formed the bottom* 64% *(Obbard et al., 2016). The* 1.89 *m Iceberg Site ice core was located at* 77°7.131′ *S and* 164°6.031′ *E and had a temperature gradient ranging from* −17.7 °*C at the top to* −2.3 °*C at the bottom. Relative to the Butter Point core, the Iceberg Site core had more frazil ice (0 cm to 30 cm), more columnar ice (30 cm to 137 cm) and less platelet ice (137 cm to 189 cm) (Obbard et al., 2016).*

**2. Were the cores sampled on site or after transportation? If the latter then please give details of how the cores were treated between extraction and sampling.**

Cores were imaged after transportation and storaged in a 33 °C cold room. We have added the following sentences to the first paragraph of the methods section in the revised manuscript:

*Immediately following core extraction, we recorded the temperature profile at* 10*-cm intervals, and stored the cores in a* 20 °*C freezer at McMurdo station prior to shipping. We then transported the cores at a constant temperature of* 20 °*C back to Thayer School of Engineerings Ice Research Laboratory at Dartmouth College, and stored them in a* 33 °*C cold room prior to analysis.*

**3. In section 3, with reference to Figures 2 and 3, please explain what the different parameters plotted are, what you might expect them to show, and how they relate to each other. For example, what is expected to be the relationship between brine volume fraction and specific surface area (Spor). In investigating permeability in sandstone samples Zhang and Weller (2014)\* have demonstrated that there is a relationship between fractal dimension and Spor. Would any such relationship be expected here? Explain the Euler number for those unfamiliar with it. How is the degree of anisotropy derived? Given that the lower parts of the cores are explained to have a significant degree of uncertainty associated with**

the measurements (quantify this?) does it make sense for the scales for Euler number and connectivity to be dictated by the lowermost sampes? *Zhang & Weller, 2014. Geophysics, 79, D377-D387. The above comments are in fact relatively introductory and indicate the need for a clear explanation of the background to the study, how the initial sampling was carried out, and what the initial measurements show. Unfortunately I find that from this point onwards the manuscript becomes confusing and is not at all well explained or illustrated for the more general reader.**

In the revised manuscript we decided to remove the Euler number, connectivity, and fractal dimension. Thus, we were able to condense Fig. 2 and Fig. 3 into a single figure in the revised manuscript. Additionally, although full descriptions of all metrics were referenced (Lieb-Lappen et al., 2017), we revised the second paragraph of the results section to include clearer explanation of these metrics in this manuscript as well. This paragraph now reads as follows in the revised manuscript:

*Since the cooling stage did not significantly warm samples beyond $-7\,^{\circ}C$, we were not surprised that general trends shown in Fig. 2 for all metrics did not differ significantly from the same samples scanned isothermally and presented in Lieb-Lappen et al. (2017) as the percolation threshold was not crossed. As in Lieb-Lappen et al. (2017), we used the structure model index $(SMI = 6\left(\frac{S'\times V}{S^2}\right)$, where $S'$ is the derivative of the change in surface area after a one pixel dilation, $V$ is the initial volume, and $S^2$ is the initial surface area) to quantify the similarity of the brine phase to plates, rods, or spheres. To quantify size, we calculated a structure thickness by first identifying the medial axes of all brine structures and then fit the largest possible sphere at all points along said axes. The structure thickness is defined as the mean diameter of all spheres over the entire volume. The structure separation is the inverse metric, providing a measurement on the spacing between individual objects. We then calculated the degree of anisotropy by finding the mean intercept length for a large number of line directions, and forming an ellipsoid with boundaries defined by these lengths. The eigenvalues for the matrix defining this ellipsoid are calculated, and correspond to the lengths of the semi-major and semi-minor axes. The ratio of the largest to smallest eigenvalues then provides a metric for the degree of anisotropy, with 0 representing a perfectly isotropic object and 1 representing a completely anisotropic*

*object. We observed that the brine phase specific surface area increased with depth, structure model index was roughly 3 (indicative of cylindrical objects), structure thickness decreased, structure separation increased, and the degree of anisotropy increased thoughout the middle of the core (Fig. 2). From the metrics above, we conclude that brine channels are primarily cylindrical in shape with more branches at lower depths, consistent with previous observations (Lieb-Lappen et al., 2017).*

**4. I find section 4.2 extremely confusing, including the figures that accompany it. Figures 5 and 6 ostensibly show the 5 "largest" brine channels from each of the two cores. I assume that one row represents 1 brine channel shown as a sequence of samples through the core in depth order. However, the colour scale suggests that the throat size goes to zero at many points i.e. a pore terminates. I find this hard to understand in the context of the fact that a single row represents a "large" brine channel. Please clarify. The discussion on branches, with reference to Figure 7, is similarly very confusing. In Figure 7 what is the horizontal scale for each separate line representing a brine channel? What is the significance of the fact that one brine channel appears to pass almost right through a sample but the others do not? I cannot at all understand the significance of Figures 8 and 9.**

We thank Referee #2 for noting this potentially confusing section. In fact, there is not a connection from the 5 brine channels selected at one depth to the 5 brine channels selected at the next depth. The reasoning is that we did not scan the entire length of the ice core. Even if we had, we would not have been able to fully track a brine channel from one sample to the next. We have added the sentence below alerting readers of this fact to avoid further confusion. Referee #2 is correct in observing that many of the selected brine channels do terminate and do not continue to the next depth. Further, we have removed the confusing Figure 7 from the revised manuscript in part due to the suggestion in the next comment and in part because it is not essential to the presented work. Fig. 8 and 9 (now Fig. 6 and 7 in the revised manuscript) are directed to the visual learners to illustrate both the magnitude and distribution of throat sizes. We have previously revised the captions to these Figures to make them clearer in response to Referee #1.

*We note that since the entire length of the cores were not scanned, there is no correlation between the five brine channels selected from one subsample (e.g. 20-cm depth) to to the next (e.g. 30-cm depth).*

**5. Similar comments apply to sections 4.3, 4.4 and 4.5, and their associated Figures. An excellent example of network terminology that will be unfamiliar to most is the statement ". . .treated the network as a directed graph. . ." etc. In essence therefore, although I believe that the work presented is ultimately publishable, to be so requires considerable restructuring of the manuscript and I urge the authors to do this. There needs to be a much clearer explanation of the techniques, probably a reduction in the number of figures including clear explanations of what they represent. A somewhat broader review of previous work on looking at the interconnectness of brine channels in sea ice would also not be remiss.**

In the revised manuscript we have completely rewritten the methods section and have provided better description of the network terminology. We feel as though this will help clarify the confusion in sections 4.3, 4.4, and 4.5. However, we have also added a reminder to the reader in section 4.5 that a directed graph in our model is one that allows fluid to flow downwards from node to node but not upwards. Throughout these sections, we have reduced the use of network terminology and/or provided better explanations in these instances. In regards to the number of figures, we have cut Fig. 3 and Fig. 7 from the original submission. Finally, we have added the following paragraph to the introduction in the revised manuscript.

*Network models have successfully been employed in a variety of fields to describe complex phenomena and predict future behaviour, in particular fluid flow in porous media (e.g., Golden et al. , 1997, Berkowitz and Balberg, 1992, and Fatt, 1956). Specific to sea ice, Freitag (1999) utilized a Lattice-Boltzmann model to model fluid flow through sea ice. Meanwhile, Golden (1998) examined critical percolation thresholds in their network model of sea ice. More recently, Zhu et al. (2006) used a two-dimensional pipe network model to simulate fluid flow through sea ice using a fast multigrid method. This network compared well with lab data for porosity above 0.15, but overestimated permeability at lower porosities Golden (Golden2007). The majority of these models generate connectivity networks based on bulk brine proper-*

*ties. Here we derive finer-grained statistics empirically, allowing for models to more closely align with the physical properties of sea ice.*

---

## Author Response (AR2)

**RESPONSES TO EDITOR**

**General Comments:**

**The key results presented in this paper are detailed microphysical analysis of two cores. You are developing some new statistical metrics to describe connectivity in brine channel networks. As the paper is written it is very difficult to follow how this builds a model as claimed in the abstract. I can see how particular parameters provide limits on fluid flow through the ice, and I think this is what you are getting at in your concept of a statistical model. The manuscript could be greatly improved if you clarify the model in a summary near the end of the paper. Specifically, the sentence in your conclusion "The probability distributions shown represent a sampling of the various microstructural behaviors needed to statistically create a brine channel network parameterized by depth and temperature" is really overly general. Can you follow on from this statement to lead the reader through the structural behavior that characterizes the network? What does this structural behavior mean for physical processes controlling fluid flow in the ice A guiding paragraph would really help lead the reader to why your rather detailed results sections are worth delving into.**

First, we thank the Editor for her detailed review and help in making this manuscript more understandable to the Cryosphere readership. We have completely rewritten the conclusion section in the revised manuscript following the sentence identified. We feel this gives more concrete examples of the structural behavior we observed with our model and what are its implications. The new conclusion section addresses this and later comments from the Editor and referees in regards to physical interpretations.

**There are still places in your methodology where it becomes hard to follow how you build particular metrics. My confusion is around visualizing the subsamples, and how pores are tracked between layers in the core. I would suggest that you consider if you can use terminology that is more intuitive for the reader to understand. I am happy with your mathematical description provided I can find my way through, and some readers may get confused by terminol-**

ogy that is not intuitive to them!

We feel as some of the confusion might have been a result of the misuse of the word "subsample." We have removed all instances of that word and have better clarified what we mean by "sample." Additionally, we have color-coded Fig. 1 that helps illustrate our definition of a brine channel. We feel as though this should alleviate the confusion in our description.

**Your observations paint a picture of the differing structure of brine channals in columner and granular ice, and how these vary with depth and temperature in a core. While this is technically interesting, and I can see the application to building models of brine in sea ice, it is not clear from your manuscript why the reader should really care about these observations. What is the broader context? Can you provide some statements as to the impact of your findings in the conclusions? You do not describe how the metrics developed can be used to characterise sea ice properties that are relevent to processes within or associated with sea ice. This broader context is needed for this paper to be impactful and meet the standards of The Cryosphere. I echo reviewer 1's comment (which they made in regard to the discussion on Markov chains) that more physical interpretation would strengthen the paper.**

The new conclusion section in the revised manuscript addresses these concerns and questions.

**Minor comments. Line numbers refer to the manuscript with markup.**

**page 4, line 15: Why does the maximal ball method best fit the microCT data? Are their trade offs in accuracy using this method compared to others?**

There is actually not much of a tradeoff since the brine channels are mostly convex. We chose the maximal ball method as it is the least computationally intensive. We have edited this sentence to better reflect our reasoning for this choice and it reads as follows in the revised manuscript:

*While other methods exist – e.g., random pipe, medial axis, and flow*

*velocity methods (Zhu et al., 2006; Dwyer, 1993; Silin and Patzek, 2006; Dong et al., 2008) – the various methods would yield similar results since the brine channels are mostly convex, and our choice of method is the least computationally intensive.*

**page 4, line 16: Missing "a" after "structure in"**

We have made the recommended revision in the revised manuscript.

**page 4, line 26: Would it be clearer to the reader to not use "our" in "the direction of our progression through the sample"? I believe you are moving downward through core slices in your analysis, which could be though of as following fluid pathways in gravity drainage. Is this correct? Perhaps rephrase to refer to the sample order more precisely.**

We have edited this sentence and it now reads as follows in the revised manuscript:

*In the language of networks (Newman, 2011), this is a directed edge as it points in a particular direction which, in this case, is vertically downward.*

**page 7, line 21: "is" missing from before "shown in Fig. 3".**

We cut the identified sentence from the revised manuscript as it was not needed. We edited the first sentence of the paragraph to include the reference to Fig. 3. The first two sentences of the identified paragraph read as follows in the revised manuscript:

*We compared the $\mu CT$-measured brine volume fraction to expected values derived from the Frankenstein and Garner relationship relating temperature, salinity, and brine volume fraction (Frankenstein and garner, 1967; Cox and Weeks, 1983) in Fig. 3. For this analysis, we used the core temperatures measured in the field and salinity values estimated from ion chromatography measured chloride concentrations presented in Lieb-Lappen and Obbard. (2015).*

**page 8, line 23: It is not clear what the "subsample" is. Can you refer back to the methodology section here, as I do not rememver**

**subsample being defined.**

We thank the editor for catching this error in the manuscript. We had previously used the terms "subsample" and sample interchangeably. In the revised manuscript we have removed all instances have "subsample" and only use the term "sample". Additionally, we added the following sentence to the first paragraph of the methods"

*We will use the term "sample" throughout this manuscript to refer to a particular 1-cm cube from a specified depth.*

**In general it may help the reader if you can more succiently allign your description of connectivity with actual brine channels in figure 1, perhaps leading readers through an example of how a brine channel is followed through the core.**

We have edited Fig. 1 in the revised manuscript to have different colors for the different brine channels. We then refer to these colors in the identified paragraph and the following to help the reader visualize a single brine channel. We note that there is no connection from brine channels in one sample to brine channels in another sample at a different depth even though it is the same core. This is noted in the second sentence of section 4.2.

**page 11, line 6: Two "boths", one needs to be removed.**

We have removed the first "both" in the identified sentence.

**page 12, line 20: "we could selected the 15316 paths" - strange grammar. I would suggest you carefully proof read, there are a few places where grammar could be improved.**

We have edited the identified sentence and the previous one in the revised manuscript, which now reads as follows:

*Summing over all brine channels in the Butter Point core resulted in 63 763 directed paths, of which 15 316 paths had a length of at least 50 steps (750 $\mu m$). We then used this smaller subset for statistical analysis of minimum throat size ($r^{min}$), maximum throat size ($r^{max}$), and summed throat size.*

**page 12, line 22: Statistical repeated in this sentence, and seams redundant.**

We have removed the second "statistical" in the revised manuscript.

**page 13, line 23-24: Many readers will skip most of the paper and just read the conclusions. They will not know what "throat size" is. The paper would be much stronger if you can write the conclusions in plain language, pointing out what your findings actually mean for fluid flow in sea ice, for example.**

We have completely rewritten the conclusion section in the revised manuscript, removing references to complex terminology such as "throat size." We have added a couple of paragraphs to point out examples that will aid the casual reader in undertanding the significance of this manuscript.

**Caption figure 2: "the Iceberg Site". Correct throughout.**

We have inserted the word "the" prior to "Iceberg Site" throughout the entire revised manuscript.

**Caption figure 4: I think you need to replace "sample" with "sub-sample".**

As noted previosuly, we have removed all instances of "subsample in the revised manuscript.

**Ditto for figure 5.**

As noted previosuly, we have removed all instances of "subsample in the revised manuscript.

[revised manuscript text omitted]

**5    Conclusions**

The primary objective of this  work has been to improve our characterization of brine channel topology, morphology, and connectivity, ~~thereby providing the statistical framework that we can use to create a sea ice brine channel network model. Since there was roughly a linear relationship between sample temperature and depth in the ice core, trends observed with one variable could easily be converted to the other. Here we have presented quantitative metrics of the brine channel topological complexity, degree of connectivity, average and individual throat sizes, probability distributions for branches to split and join, capacity for fluid flow. The probability distributions shown represent a sampling of the various microstructural behaviours needed to statistically create a brine channel network parametrized by depth and temperature.~~ in order to provide sea ice modelers with a greater level of detail on the factors that affect microstructural transport properties. While most percolation models use coarse microstructural properties to form a statistical basis for predicting connectivity, ours derives finer-grained statistics empirically, allowing for better representation of the range of physical properties found in sea ice of different types and conditions. We can statistically model the evolution of brine channels as we move downwards through the sea ice cover. Beginning with an initial brine pocket, our estimates of the evolution probability distributions from $\mu$CT scans of sea ice samples tell us how the channel changes as we progress downward through the sample - Does it grow? Shrink? Split into more than one branch? Join up with more than one branch? Close off entirely?

Overall, we observed similar morphological profiles for both first-year sea ice cores.  Topological complexity had the expected c-shape profile  that is consistent with complex frazil ice in the top of the core, relatively cold columnar ice below it, and increasingly warmer columnar ice at lower depths.  However, we did not have good success in imaging and thresholding  ice with the warmest in-situ temperatures at the bottom of the core.

Our estimates of the evolution probability distributions provides a stochastic model of brine channels within sea ice at different temperatures, extending the percolation models described above. Their structural features reveal the onset of transitions between different types of ice: in our analysis, we see different statistical features that delineate frazil and columnar ice. Further, the level of detail inherent to this technique allows us to quantify some of the finer details of brine channel structure and development. In addition to estimating the expected brine volume and permeability for ice at a fixed temperature, we can see when and why permeability arises by analyzing the probabilistics structures. For example, Fig. 8 shows a stark structural difference between frazil and columnar ice which points to the onset of percolation: brine pockets in frazil ice larger than about 1 mm are extremely likely to join or split while the largest brine pockets in columnar ice are more likely to persist. We observe that brine channels in columnar ice simply continue downwards with little change in  size.

For those in which there is a significant change in  size, there is a likely fork leading to a split in the channel.  A higher probability of interconnections between brine channels translate directly into a higher probability of permeability of the ice.

5    When examining the branching in brine channels, we  observe that the largest channels  have the greatest number of branches, but overall  brine channel size does not appear to have a direct correlation with the number of branches.  Brine channel size is most dependent upon the depth and consequently ice type. When a split in a brine channel does occur, it is most likely to split into two child branches, and after the split, the brine generally has access to a larger region of the sea ice than before. Starting and ending  brine pocket sizes are strongly correlated with the

10   flow capacity with larger initial/final  sizes strongly indicative of increased flow. We detected pinch points in the brine channels that are critical points when determining whether the sea ice cover has crossed the percolation threshold. However, further work is needed in examining warmer ice with greater brine volume fractions.

Our framework enables us to statistically replicate the pore structure of sea ice at different depths and temperatures. The next step for this work is to create a brine channel network from the  probability distributions presented here. For a sample

15   at a given depth/temperature, first an initial  region at the top  of the ice would be selected  with size consistent with the  statistics shown here.  The brine channel 
[revised manuscript text omitted]